# Identification of Small Regions of Overlap from Copy Number Variable Regions in Patients with Hypospadias

**DOI:** 10.3390/ijms23084246

**Published:** 2022-04-12

**Authors:** Carter H. Scott, Ina E. Amarillo

**Affiliations:** 1Washington University School of Medicine, St. Louis, MO 63110, USA; carter.scott@wustl.edu; 2Department of Pathology and Immunology, Washington University School of Medicine, St. Louis, MO 63110, USA

**Keywords:** hypospadias, DNA copy number variations, microarray analysis, small regions of overlap, CNV map, differences of sex development

## Abstract

Hypospadias is a common form of congenital atypical sex development that is often associated with other congenital comorbidities. Many genes have been associated with the condition, most commonly single sequence variations. Further investigations of recurrent and overlapping copy number variations (CNVs) have resulted in the identification of genes and chromosome regions associated with various conditions, including differences of sex development (DSD). In this retrospective study, we investigated the DECIPHER database, as well as an internal institutional database, to identify small recurrent CNVs among individuals with isolated and syndromic hypospadias. We further investigated these overlapping recurrent CNVs to identify 75 smallest regions of overlap (SROs) on 18 chromosomes. Some of the genes within these SROs may be considered potential candidate genes for the etiology of hypospadias and, occasionally, additional comorbid phenotypes. This study also investigates for the first time additional common phenotypes among individuals with hypospadias and overlapping CNVs. This study provides data that may aid genetic counseling and management of individuals with hypospadias, as well as improve understanding of its underlying genetic etiology and human genital development overall.

## 1. Introduction

Hypospadias is a form of congenital atypical sex development that involves a urethral meatus located on the ventral aspect of the penis, scrotum, or perineum. It is very common, with an estimated incidence of one per 125 to one per 300 live male-assigned births, which has increased in frequency over the past few decades [1]. While most cases are distal to the neck of the glans, known as anterior hypospadias, the remaining cases have more proximal meatuses, known as middle (penile) or posterior (penoscrotal, scrotal, or perineal) hypospadias. Hypospadias is in contrast to epispadias, in which the urethral meatus is found on the dorsal penis.

Hypospadias can lead to both cosmetic and functional impairment, but only when severe. Infants with hypospadias, especially posterior cases, more frequently exhibit other congenital comorbidities compared to infants born without hypospadias [2]. In addition, people born with hypospadias are often found to have other urogenital abnormalities, especially cryptorchidism (undescended testis or testes) [2,3].

Despite decades of research, the etiology of most cases is unknown, likely a mix of environmental and hormonal factors with monogenic and/or multifactorial genetic features [2,4]. Specifically, it is likely due to a mix of hypoandrogenic states combined with pre-existing genetic susceptibility [5]. Environmental factors repeatedly associated with hypospadias include maternal hypertension and pre-eclampsia, placental insufficiency, maternal intrauterine exposure to diethylstilbestrol, being small for gestational age, and low birth weight [2,4]. About 7–25% of cases occur as part of familial clustering [6,7,8], and in pedigree and twin studies, heritability has been reported as anywhere from 55 to 77% [9], suggesting that genetic factors indeed play a strong role in causing the phenotype.

Over five decades of research have revealed many genes that are associated with hypospadias, most commonly involving sequence variations [2,10,11,12,13,14,15,16,17,18] (Table 1). Although several variants have been described as potentially causing hypospadias, researchers largely agree that very few cases of isolated hypospadias—especially milder ones—are caused by single sequence variations [2]. A limited but growing body of research suggests that epigenetic changes may contribute to differences of male sexual development, including hypospadias [19].

The advent of chromosomal microarray analysis (CMA) allows for the identification of copy number variations (CNVs) across the entire genome. Further investigations of recurrent and overlapping deleted and/or duplicated regions have resulted in the identification of genes and regions associated with various conditions, including differences of sex development (DSD) [20]. These documented CNVs are typically larger in size (≥1.5 Mb) [21]. In this retrospective study, we investigated the DECIPHER database (DatabasE of genomiC varIation and Phenotype in Humans using Ensembl Resources) of deletions and duplications detected in patients with various clinical phenotypes [22], as well as an internal institutional database, to identify small recurrent CNVs among individuals with isolated or syndromic hypospadias. We further investigated these overlapping recurrent CNVs present in at least three patients to identify the smallest regions of overlap (SROs) between their CNVs, and the genes within these intervals, some of which can be considered potential candidate genes for the etiology of hypospadias. We then constructed a whole-genome CNV map from these SROs to illustrate the global genomic view of hypospadias. Furthermore, this study investigates for the first time additional common phenotypes among individuals with hypospadias and overlapping CNVs. This study expands genotype–phenotype correlation and provides data that may aid genetic counseling and management of individuals with hypospadias.

## 2. Results

The DECIPHER query returned 469 hypospadias cases, for which the number of reported copy number variants ranged from 0 to 7. About 76% (357/469) only had one reported variant, while the rest had at least two variants [22]. The majority were 46XY; however, we also found ten cases reported as 46XX, 22 reported as “other” or “unknown”, and one reported with sex chromosome aneuploidy [22].

Our CMA approach identified 75 small SROs (Figure 1) with a mean size of 136.04 kilobase pairs (kb) (range, 0.21–574.57 kb), and a mean of 8.9 individuals with overlapping CNVs (range, 3–28) (Figure 2; Table 2 and Appendix A). Thirty-one of the 75 SROs (41%) included at least one CNV of a patient from our center. While most SROs included a mixture of deletions and duplications, 7 SROs exclusively included deletions, while 8 included only duplications. SROs were found across the genome, with the highest frequencies of SROs found on chromosomes 22 (17 SROs) and 1 (9 SROs); no SROs were identified on chromosomes 3, 14, 19, 20, 21, or the Y chromosome (Figure 3).

Of the loci given for the nine hypospadias phenotype entries in OMIM^®^, we found four SROs within the locus for susceptibility to X-linked hypospadias (*HYSP4*, Xp11.22) [11]. There was only one known gene within these SROs (*HUWE1)*, which was identified as a candidate gene. We also identified SROs overlapping with the following genes with established associations with hypospadias: *RBFOX2*, *SHH*, and *WT1.*


Several studies have used other methods to identify CNVs potentially involved in hypospadias, although they typically use smaller sample sizes and do not discriminate based on CNV size. Such studies have associated duplications of 17q12 [23], as well as duplications of Ypter-Yq11.223 and loss of the remainder of chromosome Y (mosaic dicentric Y) [21], but neither overlapped with our SROs. Similarly, we did not find any SROs overlapping potential hypospadias susceptibility loci identified on chromosomes 2, 7 9, and 10 in two studies using familial linkage analysis [24,25]. However, our findings support one study which described clinically significant CNVs on chromosomes 2, 5, 12, 16, and X [26]: we identified 3 SROs within their CNV at 16p11.

### 2.1. Candidate Genes

These SROs involved a total of 242 genes; 20 genes had an HI score ≤10%, and 36 had a pLI score ≥0.9 [22]. A total of 43 genes met either of the criteria. Of these 43 genes, 40 (93%) had never been identified as being associated with hypospadias and were thus identified as candidate genes (Table 2). DECIPHER does not provide triplosensitivity data, but we expanded our analysis to include this data. Of the candidate genes, they either had no evidence or record of triplosensitivity. Although the Xp11.22 region (SRO070-073) is documented to be triplosensitive, the candidate gene *HUWE1* within SRO073 is not [27]. Of the 40 candidate genes, 30 (75%) had low tissue specificity; the only gene with specificity for male tissues was *EHF*, which is tissue enhanced in ductus deferens, seminal vesicle, and salivary gland [28]. Thirteen of the 40 candidates (33%) had a “high” protein expression score in at least one of the male tissues for which protein expression scores are available (epididymis, prostate, seminal vesicle, testis) [28] (Appendix A).

Through literature review, we identified 80 genes previously published in association with hypospadias as either being causative, a risk factor, or associated with isolated hypospadias in screening studies, as well as genes proposed as candidates in gene expression studies or given their role in genital and gonadal development (Appendix A). The majority of these are sequence variations. Additionally, more than 600 genes were listed in OMIM^®^ as having at least one patient with hypospadias as a reported phenotype [29].

We also identified 14 recurrent copy number regions with sizes ranging from 0.21 kb to 186.74 kb. These regions are gene deserts, or regions with no protein-coding genes.

### 2.2. Common Comorbidities

Of the 469 DECIPHER cases with hypospadias, only two had no documented comorbidities (“isolated hypospadias”), and both only had one reported CNV. Of note, one of these CNVs is the basis for SRO041, which is 30.67 kb narrower than the CNV; this SRO contained 2 candidate genes, *CYFIP1* and *NIPA2*. The remaining 467 DECIPHER cases had anywhere from 2 to several dozen comorbid phenotypes affecting nearly all organ systems [22]. The most commonly reported comorbidity present in at least three cases with CNVs overlapping an SRO was neurodevelopmental abnormality, including intellectual disability and neurodevelopmental delays such as global developmental delay or delayed speech and language development (found in 24 SROs). Other abnormalities of the genitourinary system were found, including micropenis (3 SROs), cryptorchidism (1), and hydronephrosis (1). Comorbidities found in multiple (2 to 4) SROs include short stature, small size for gestational age, inguinal hernia, feeding difficulties in infancy, microcephaly, hypertelorism, low-set ears, hearing impairment, abnormality of the pinna, anteverted nares, micrognathia, and cleft palate. For other comorbidities and their proportions, please reference Appendix A.

## 3. Discussion

Human genital development is complex, controlled by an intricate network of factors regulating endocrine function, organ development, and sex determination and differentiation. The fragility of this network is demonstrated by the high incidence of DSD worldwide, estimated at 1:4500 to 1:5000 live births [30]. Patients with isolated hypospadias, even when severe, are typically not referred for genetic testing. When initial hormonal analyses are unremarkable, the abnormality is usually deemed an isolated anatomical atypia [31,32]. Therefore, the causes underlying most cases have not been found. Indeed, there were only two patients with isolated hypospadias in the entire DECIPHER database. In this study, chromosome microarray-based technology (CMA) identified many copy number variations recurrently found in individuals with hypospadias, and sometimes with other phenotypes as well. Although some genes have been implicated, the genomic etiology of hypospadias remains poorly understood. Many genomic variants of individuals with hypospadias are exceedingly rare, making genotype-phenotype correlations uncertain and thus clinical interpretation challenging. While a small number of CNVs have been identified in individuals with hypospadias, they have either focused on large (>1.5 Mb) CNVs or studied few (<30) participants [21,26]. In fact, few CMA studies have investigated small (<1 Mb) CNVs in any human genetic disorder [33]. Thus, the contribution of CNVs, especially those with few genes, to the etiology of hypospadias remains largely unexplored.

The use of large, multi-institutional databases allows for the identification of individuals that share both phenotypic features and gene variants, which can improve certainty in gene pathogenicity and may allow for clarification of the role of novel genes in development, physiology, and disease. The largest of such databases to date is DECIPHER, which compiles genotype and phenotype information of more than 35,000 individuals from over 250 centers around the world [22].

Use of these large-scale databases can allow for the identification of small CNVs recurrent among database participants with certain phenotypes. Historically, CNVs reported in association with syndromes have been large, containing many genes. However, the advent of publicly available, high-resolution CNV maps (with CNVs as small as 50 bp) now allows for comparison of overlapping CNVs of various sizes to identify SROs, or the smallest common CNV associated with a specific phenotype [34,35]. More recently, SROs as small as 5.2 kb have been identified upstream of *SOX9* in patients with DSD [20]. These data, then, provide an opportunity to construct maps of these SROs across the genome. Rather than the large, nonspecific CNVs previously reported with various genetic syndromes, these small CNVs sometimes contain one or few genes, intragenic regions, and regulatory regions. Importantly, these maps may assist with shrinking the genomic “gap” in understanding human genetic disorders, especially phenotypes with complex etiologies such as hypospadias.

Our findings illuminate for the first time many potential candidate genes not previously thought to play a role in hypospadias etiology, as well as many chromosomal structural imbalances that may serve as risk factors for this and other phenotypes. Additionally, these results demonstrate the powerful potential of chromosome microarray-based analysis in the discovery of genetic factors contributing to diseases with complex etiologies.

### 3.1. Hypospadias and Sex Chromosomes

It might be assumed that all cases of hypospadias are exhibited by Y chromosome-bearing patients (e.g., XY, XXY). However, DECIPHER had several cases reported as 46XX and even more reported as “other” or “unknown.” Given that hypospadias may occur in the spectrum of DSD, it is imperative that sex chromosomes are reported and analyzed when performing research on this and other conditions related to DSD [36]. Since the major classification of DSD is based on sex chromosomes, it may be helpful to include the reason why sex chromosomes are listed as “other” or “unknown.”

The Y chromosome has long been speculated to be associated with hypospadias; some of the earliest research on the genetics underlying the condition described several cases with a variety of translocations and deletions on this chromosome [37]. However, the lack of SROs found on the Y chromosome support more recent research that has found no CNVs on this chromosome, including *SRY*, in patients with isolated hypospadias [38,39]. The lack of SROs on this chromosome and chromosome 2 are also in agreement with a paper that found no recurrent and overlapping CNVs in patients with DSD on either of these chromosomes [33].

### 3.2. SRO Relationships to Previously Described Regions and Genes

Our study found very few SROs that overlapped with previously described regions or genes. This is in agreement with the fact that existing studies have largely associated cases of hypospadias with sequence variations, rather than CNVs [2] (Appendix A). However, this may change given the recent increase in CMA utilization, use of gene-targeted CMA, increased data sharing through DECIPHER, or increased genetic evaluation of patients with hypospadias.

### 3.3. Candidate Genes

One study used DECIPHER to identify recurrent CNVs in patients with isolated (“non-syndromic”) congenital genitourinary anomalies [40]. That study identified *RBFOX2* as a candidate gene and used mouse models to explore its role in upper and lower genitourinary tract development. Our study used a similar approach to identify SROs; however, it explored the whole genome and utilized the CNVs of patients with both syndromic and non-syndromic hypospadias. Through these methods, we identified many regions and genes that have never been reported as being associated or potentially associated with hypospadias. 

Certain candidates have high tissue specificity, both within and outside of the male urogenital system [12] (Appendix A). Some SROs containing these candidates had common comorbidities related to the extra-urogenital tissues in which they are enhanced. For example, *DLGAP1* (SRO052) and *GRIN2A* (SRO048) are tissue enriched in the brain, and the majority of cases within these SROs had neurodevelopmental abnormalities. Additionally, *PTPRD* is tissue enhanced in the brain, and 7 of 11 cases in the SRO (SRO030) were also reported to have intellectual disability, and 4 cases had microcephaly.

Many candidate genes with tissue specificity did not have recurrent phenotypes in this study population, such as *RBM8A* (SRO002), which is tissue enriched in blood. However, more in-depth investigation of these tissues in individuals with relevant CNVs may identify common comorbidities. Additionally, investigation of cases with hypospadias and variants in genes enriched in multiple tissues (e.g., *ADAMTSL1*, *BDNF)* may potentially lead to the identification of new syndromes, or the expansion of phenotypes for existing syndromes.

Of the 40 candidate genes, it would seem prudent to most strongly consider those candidate genes with high specificity and expression in “male tissues” (e.g., *EHF*, *EP400*, *RBM8A*, *SNED1*, *YWHAE*, *ZC3H18).* However, that should not discount those genes expressed at low levels in male tissues, which may demonstrate important temporal expression. Such candidate genes for which the SRO is composed mostly of duplications (e.g., *DBN1*) may suggest that overexpression disrupts typical urogenital development, and those with mostly deletions (e.g., *ADAMTSL1*, *BDNF*, *COL4A1)* may suggest haploinsufficiency.

For candidate genes, further investigation of tissues in which they are highly expressed—and potentially associated phenotypes—may allow for the identification of common comorbidities (or, potentially, novel syndromes) associated with certain CNVs. For example, *DLGAP1* and *GRIN2A* are enriched in brain tissue, and 12 of the 13 cases with relevant CNVs feature neurodevelopmental abnormalities. Although cases with CNVs overlapping *PTPRD* (enhanced in parathyroid tissue) were not explicitly reported to have parathyroid-related abnormalities, further investigation may uncover relevant comorbidities. The same could be said for cases with CNVs affecting *RBM8A* (enriched in blood) and hematologic disorders. Additionally, data regarding co-expression, physical interactions, predicted co-expression, genetic interactions, co-localization, and shared protein domains may provide important clues for further functional study [41].

### 3.4. Syndromes

SROs were found within loci known to be relevant to syndromes that have previously been associated with hypospadias. Identification of SROs within these loci—and especially candidate genes—may provide insight into the specific genetic etiology of hypospadias for that syndrome. For example, Hunter–McAlpine Syndrome is associated with hypospadias, as well as other phenotypes such as intellectual disability, microcephaly, and short stature. The syndrome is known to be caused by a partial duplication of 5q35-qter [42], although the specific region of this duplication underlying hypospadias has never been proposed. SRO018 is within this locus and is composed of only cases with duplications; this SRO contains candidate genes *DBN1* and *NSD1*, which may contribute to hypospadias etiology in these cases. SROs with candidate genes were found in the established risk loci for other syndromes known to be associated with hypospadias, such as Silver–Russell Syndrome (*PSMD13*); 13q Deletion Syndrome *(COL4A1);* and 22q Deletion/Duplication Syndromes, which includes DiGeorge Syndrome (*CRKL*).

In addition, several SROs were located within known susceptibility regions for genetic syndromes for which hypospadias is not a commonly associated phenotype. These findings suggest that hypospadias may be investigated further as a potential syndromic feature, and that individuals with relevant CNVs and hypospadias may need to be evaluated further for comorbidities associated with these syndromes. 

For example, SRO041 overlaps with the newly established Burnside–Butler Syndrome, which is associated with various developmental and psychiatric disorders [43]. Notably, three of four cases included in this SRO have delayed speech and language development. This SRO contains two candidate genes, *CYFIP1* and *NIPA2*, both of which are highly expressed in ductus deferens, testis, epididymis, seminal vesicles, and prostate [28]. Both genes have been implicated in Burnside–Butler Syndrome, although hypospadias has not yet been identified as a comorbidity. Additional examples of such syndromes and SROs can be found in Appendix A.

### 3.5. Common Comorbidities

For almost 50 years, hypospadias has been associated with an increased risk of other congenital comorbidities, especially cryptorchidism [37]. One study suggests that as high as 29.4% of cases have additional comorbidities—often urogenital, but also extra-urogenital [3]. Genitourinary comorbidities were indeed common in our population, found as a recurrent phenotype in 5 SROs, although other common comorbidities affected a variety of organ systems. Another study found that individuals with hypospadias are more likely to be diagnosed with intellectual disability, autism spectrum disorder, and other behavioral or emotional disorders [44]. Although they did not provide a conclusive explanation for these findings, the increased risk persisted after adjusting for known genetic syndromes, suggesting a complex model of heritability that may be confounded by environmental, psychosocial, and endocrine factors. However, this potentially suggests that there may be unknown genetic syndromes involving both hypospadias and neurodevelopmental abnormalities. Our study found similar results, as 24 of the 75 SROs (32%) had neurodevelopmental abnormality as a phenotype present in at least three cases contributing to the SRO.

One paper found that 28% of the patients they studied—those with both hypospadias and neurodevelopmental delay—also had cardiovascular abnormalities [45]. They identified various syndromes associated with similar constellations of comorbidities, citing several pathogenically implicated genes as modulators of transcription and epigenetic regulation that may serve related functions in development. Indeed, our methods identified three SROs featuring neurodevelopmental delay and cardiovascular abnormalities as shared comorbidities. SRO008 contains *SMYD3*, a transcriptional regulator, and SRO030 contains candidate gene *PTPRD,* which plays an important role in memory, learning, and synaptic plasticity [29]. SRO023 contains *DPP6,* which is associated with hereditary ventricular fibrillation, as well as autosomal dominant (AD) microcephaly and neurodevelopmental delay which was replicated on knockdown of *DPP6* in mice [29]. A CNV analysis of patients with AD microcephaly identified two small de novo deletions in the gene [46]; which is supported by SRO023, which is composed of 13 deletions and only 3 duplications. Such findings suggest a potential regulatory role for *DPP6* in cardiac, neurological, and genital development.

### 3.6. Recommendations

Our findings provide further support for genomic influences in the etiology of hypospadias. Hypospadias is thought to arise during external genitalia development, between gestational weeks 8–12, due to incomplete fusion of the labioscrotal folds [38]. The roles of these genes in the development of hypospadias will be elucidated by studying spatial and temporal expression during this development of the urethra and nearby structures. These findings, in concert with existing genetic and environmental studies, the DECIPHER database, and animal models [47], will facilitate the functional studies needed to understand the roles of these genes in the genetic etiology of hypospadias, as well as in the broader context of urogenital development. 

In addition, these findings call for more frequent genomic analysis in patients found to have hypospadias, which may further elucidate candidate genes for this condition as well as novel genes involved in human genital development. Ideally, this analysis should include both sequencing and genome-wide CNV analyses. Additionally, the specific category of hypospadias (e.g., coronal, penoscrotal) should be reported in public databases to improve genotype-phenotype correlation. Although increased hypospadias severity has been found to be associated with increased likelihood of identifying likely pathogenic variants, patients with even mild hypospadias can benefit from genetic testing, which may provide early diagnoses, reveal undiagnosed syndromes, or identify candidate genes or variants [48]. Indeed, these results echo recommendations from panels of international experts for the appropriate diagnostic approach to DSD. These recommendations highlight the importance of increased genetic testing and results sharing in centralized databases such as DECIPHER to promote functional studies that allow for the identification of novel implicated genes and variants [49].

As our understanding of comorbidities associated with different structural abnormalities unfolds, CMA may allow for earlier screening and detection. For example, it has been suggested that *WT1* testing is not indicated in isolated hypospadias without cryptorchidism, but there have been several reports of patients with such phenotype who later develop Wilms tumors [31]. Recent research suggests that although genetic evaluation of patients with proximal (severe) hypospadias is not the typical standard of care, when evaluated, many are found to harbor clinically relevant genetic variations, supporting increased genetic evaluation of these individuals [26,32]. In addition, as the understanding of the genetic etiology of this phenotype improves, genetic counseling for families may better inform estimates of recurrence risk for subsequent pregnancies. Improved understanding of comorbidities may also allow for the provision of anticipatory guidance to parents, and thus earlier detection and treatment.

## 4. Materials and Methods

### 4.1. CNV Data Sources

DECIPHER was queried to identify all cases with “hypospadias” as a listed phenotype [22], which included those with the HPO term HP:0000047 “hypospadias” as well as HPO terms further defining hypospadias as penile, penoscrotal, glandular, perineal, midshaft, scrotal, or coronal. All cases were included, regardless of karyotype. CMA data were also obtained from 32 patients with hypospadias referred to our center since 2008. CMA data for patients from our center were acquired following the methods previously described in “Integrated Small Copy Number Variations and Epigenome Maps of Disorders of Sex Development” [33]. CMA data from these 32 cases had not yet been submitted to DECIPHER. 

To construct the whole-genome CNV map described in this study, we used the karyotype map function of DECIPHER to identify chromosome regions with three or more recurrent and overlapping deletions or duplications. From such regions, the smallest region of overlap (SRO) is defined (Appendix A). We then integrated the CNV data obtained from our center. Only SROs less than 1 Mb in size were included in this analysis.

Other genomic information collected for each SRO includes size, base-pair coordinates, chromosome locus, and gene content. We analyzed all genes within the SROs using DECIPHER’s predictive tools such as haploinsufficiency index (HI) (scores of ≤10% indicate a high likelihood of exhibiting haploinsufficiency) [50] and pLI (score of ≥0.9 indicates very low tolerance to loss of function mutations) [28]. In accordance with prior studies using these cutoff values [28,50], genes that met either of these criteria and had never been previously described in association with hypospadias were deemed candidate genes. In addition to genotype information, the spectrum of phenotypes in patients with CNVs included in the SRO was also analyzed. These SROs were cross-referenced with loci listed for the nine hypospadias phenotype entries in OMIM^®^ as well as previously described CNVs associated with hypospadias [11]. 

### 4.2. Limitations

Our analysis of potential common phenotypes was limited by how cases were entered in DECIPHER and our genetic database, including which phenotypes were reported. For example, depending on when participants’ phenotypes were last updated, or their age of death, age-related phenotypes (e.g., malignancy, short stature, intellectual disability, behavioral/emotional disorders) may not have yet manifested. In addition, distal hypospadias may not even be clinically recognized, much less reported in DECIPHER, so there is likely under-reporting of the phenotype in this database.

When using public databases, different array technologies used by reporting institutions may lead to differences in the reported size of CNVs that are actually identical [51]. For that reason, the coordinates of our SROs should not be considered absolute or definitive loci.

Although HPA lists tissue specificity and expression levels in several “male tissues,” they do not list values for tissues specifically associated with urethral or penile development; therefore, these values should only be regarded as an estimate. 

SROs that contain no genes may hold important regulatory regions. We identified three such SROs within what OMIM^®^ describes as a susceptibility locus for X-linked hypospadias [11], which were all located less than 3.5 Mb downstream of *DGKK*, the gene most commonly associated with hypospadias [2,52,53]. These regions may contain key regulatory elements for *DGKK* not previously understood in relation to hypospadias pathogenesis, such as those recently found for *SOX9* while using a similar method to identify SROs [54]. However, analyzing these regions is beyond the scope of this paper and should be considered an area for future study.

Finally, many candidate genes have been proposed to be associated with this phenotype, but such studies often report small numbers of cases and controls. Often, they are not able to be replicated in future studies, although this could be due to differences in study populations. Additionally, this study does not address the non-genetic factors contributing to hypospadias such as environmental, psychosocial, and endocrine factors. These studies, as well as our results, should not be interpreted as asserting causation, but rather as preliminary data for future areas of functional research to further understand the genetic etiology of hypospadias.

## 5. Conclusions

This study used chromosome microarray-based technology (CMA) to identify genomic structural variations recurrently found in individuals with hypospadias, a common congenital atypical sex development. These findings propose many of these loci and the candidate genes within them as novel potential underlying causes of hypospadias and, occasionally, additional comorbid phenotypes. However, given that the role of non-genetic factors remains to be explored, these candidate genes should be investigated with functional studies that will further determine their potential relevance to hypospadias. Recurrent losses and gains of DNA were detected on nearly all chromosomes, suggesting genome-wide CMA as an ideal assessment tool for this population. Increased genetic screening of individuals with hypospadias, as well as reporting these data to collaborative databases, will improve detection of common comorbidities as well as additional candidate genes. These results provide insight into many novel risk loci and genes for further investigation to improve understanding of the genetic etiology of this common, yet poorly understood condition, as well as human genital development overall.

## Figures and Tables

**Figure 1 ijms-23-04246-f001:**
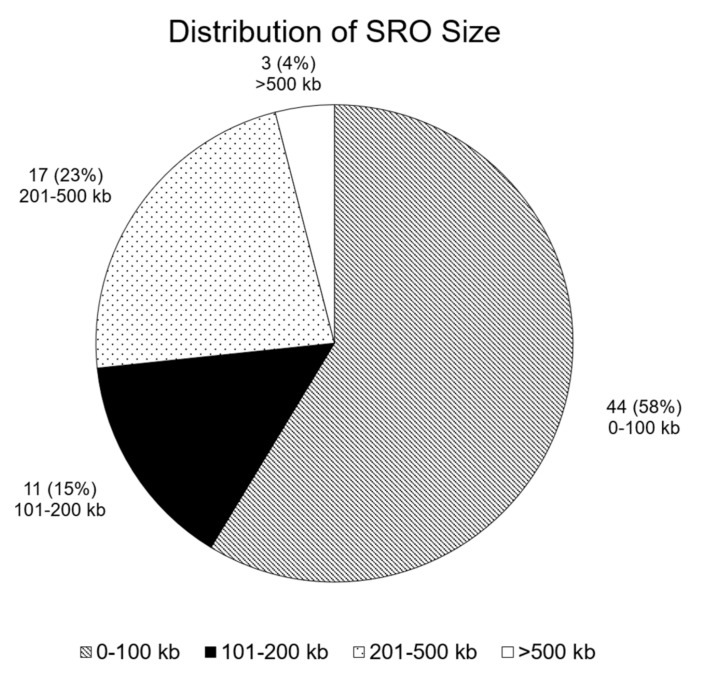
Distribution of SRO size.

**Figure 2 ijms-23-04246-f002:**
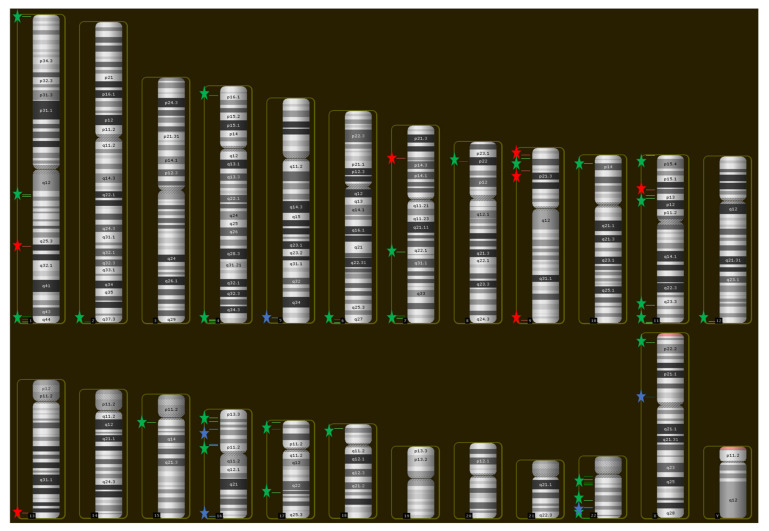
SRO Distribution across the Genome. Chromosomal locations for each SRO were used to construct a BED (Browser Extensible Data) file which was imported into Chromosome Analysis Suite (ChAS; Affymetrix, Inc./Thermo Fisher Scientific, Santa Clara, CA, USA) as a track. Green stars indicate SROs or clusters of SROs composed strictly of duplications, red stars indicate deletions, and blue stars indicate SROs that featured both types of CNVs.

**Figure 3 ijms-23-04246-f003:**
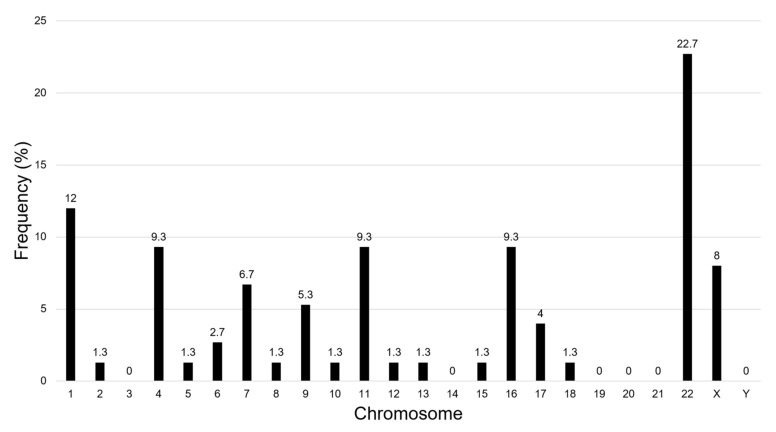
SRO frequency across chromosomes.

**Table 1 ijms-23-04246-t001:** Selected previously defined genes and regions in hypospadias.

Genes	Chromosome Locus (GRCh37/hg19)	Common Variation	SRO from Our Study (*Gene*)	Size (kb)	Reference
*LHCGR*	2p16.3	Sequence	None	N/A	Vezzoli et al. 2015 [9]
*SRD5A2*	2p23.1	Sequence	None	N/A	OMIM #607306 [10], HPA [11]
*ZEB2*	2q22.3	Sequence	None	N/A	van der Zanden et al. 2012 [2]
*HOXA13*	7p15.2	Sequence	None	N/A	OMIM #142959 [10]
*SHH*	7q36.3	Sequence	SRO024	473.21	Carmichael et al. 2013 [12]
*CYP17A1*	10q24.32	Sequence	None	N/A	Singh et al. 2018 [13]
*WT1*	11p13	Sequence	SRO036	375.54	van der Zanden et al. 2012 [2]
*CYP11A1*	15q24.1	Sequence	None	N/A	Lara-Velazquez et al. 2017 [14]
*ADAT3*	19p13.3	Sequence	None	N/A	Thomas et al. 2019 [15]
*RBFOX2*	22q12.3	Sequence	SRO063	83.35	White et al. 2020 [16]
*DGKK*	Xp11.22	Sequence	None	N/A	HPA [11]
*MID1*	Xp22.2	Sequence	None	N/A	HPA [11]
*AR*	Xq12	Sequence	None	N/A	OMIM #300633 [10]
*MAMLD1*	Xq28	Sequence	None	N/A	OMIM #300120 [10]; HPA [11]
**Regions**					
7q32.2-q36.1	-	CNV	None	N/A	OMIM #146450 [10]
13q33-q34	-	CNV	None	N/A	Andresen et al. 2010 [17]
Xp11.22	-	CNV	SRO0070-74 (***HUWE1***)	1.58	OMIM #300856 [10]

**Table 2 ijms-23-04246-t002:** Candidate genes in hypospadias.

SRO from Our Study	Chromosome Locus (GRCh37/hg19)	CNV Type	Size (kb)	Genes	Most Common Comorbid Phenotype **
SRO001	1p36.33	Del/Dup	68.05	*ATAD3A*, *ATAD3B*, *ATAD3C*	
SRO002	1q21.1	Del/Dup	81.26	*ANKRD35*, *GNRHR2*, *ITGA10*, *LIX1L*, *PEX11B*,***RBM8A ****	
SRO003	1q21.1	Del/Dup	29.44	*ANKRD35*, *NUDT17*,***PIAS3 ****	
SRO004	1q21.1	Del/Dup	103.47	*POLR3C*, *RNF115*	
SRO005	1q21.1-21.2	Del/Dup	341.53	*BCL9*, *CHD1L*, *LINC00624*, *OR13Z1P*, *OR13Z2P*, *OR13Z3P*	
SRO006	1q31.1	Del	251.69	*LINC01036*	
SRO007	1q44	Del/Dup	8.65	** *AKT3* ** *****	
SRO008	1q44	Del/Dup	15.32	*SMYD3*	Ventricular septal defect
SRO009	1q44	Del/Dup	178.41	*OR2L13*, *OR2M1P*, *OR2M2*, *OR2M3*, *OR2M4*, *OR2M5*	
SRO010	2q37.3	Del/Dup	3.55	*MTERFD2*,***SNED1********	Hypotonia
SRO011	4q35.2	Del/Dup	481.63	*ADAM20P3*, *ZFP42*	Feeding difficulties in infancy
SRO012	4q35.2	Del/Dup	9.73	*-*	
SRO013	4q35.2	Del/Dup	94.01	*RNU6-173P*, *TRIML2*	Feeding difficulties in infancy
SRO014	4q35.2	Del/Dup	3.61	*-*	
SRO015	4q35.2	Del/Dup	38.47	*-*	
SRO016	4p16.1	Del/Dup	148.71	*MIR4798*	
SRO017	4p16.1	Del/Dup	68.85	*SORCS2*	Hypertelorism
SRO018	5q35.3	Dup	237.41	***DBN1********, *F12*, *GRK6*, *LMAN2*, *MXD3*,***NSD1********, *PFN3*, *PRELID1*, *PRR7*, *PRR7-AS1*, *RAB24*, *RGS14*, *RN7SL562P*, *SLC34A1*	
SRO019	6q27	Del/Dup	3.77	*C6orf123 (lncRNA)*	Hypoplastic nail
SRO020	6q27	Del/Dup	235.235	*FAM120B*, *PDCD2*, *PSMB1*, ***TBP********	
SRO021	7q22.1	Del/Dup	133.74	** *CUX1* ** *****	
SRO022	7q36.2	Del/Dup	12.67	*-*	
SRO023	7q36.2	Del/Dup	22.22	*DPP6*	Congenital heart defect
SRO024	7q36.3	Del/Dup	473.21	*SHH*	
SRO025	7p15.2	Del	112.98	*SNX10*	
SRO026	8p22	Del/Dup	88.73	*MSR1*	
SRO027	9q34.3	Del	204.21	*NACC2*, *C9orf69, LHX3*, *QSOX2*	
SRO028	9p22.2-22.1	Del	337.25	***ADAMTSL1********, *MIR3152*, *RN7SKP258*	
SRO029	9p24.1	Del	159.76	*ERMP1*, *KIAA1432*	Cryptorchidism
SRO030	9p24.1	Del/Dup	333.12	** *PTPRD* ***** **	Anteverted nares
SRO031	10p14	Del/Dup	155.6	*LINC00707*	
SRO032	11q23.3	Del/Dup	54.78	*GRIK4*	Anteverted nares
SRO033	11q25	Del/Dup	378.38	*B3GAT1*, *GLB1L2*	
SRO034	11q25	Del/Dup	186.74	*-*	Abnormality of the pinna
SRO035	11p14.1	Del	4.62	** *BDNF* ** *****	
SRO036	11p13	Del/Dup	375.54	*CCDC73*,***EIF3M********, *HNRNPA3P9*, *WT1*, *WT1-AS*	Micrognathia
SRO037	11p13	Del/Dup	112.5	** *EHF* ** *****	Short stature
SRO038	11p15.5	Del/Dup	323.55	*ATHL1*, *B4GALNT4*, *BET1L*, *CICP23*, *IFITM1*, *IFITM2*, *IFITM3*, *IFITM5*, *LINC01001*, *NLRP6*, *ODF3*, *OR4F2P*,***PSMD13********, *RIC8A*, *RNU6-447P*, *SCGB1C1*, *SIRT3*	
SRO039	12q24.33	Del/Dup	574.57	*DDX51*,***EP400********, *EP400NL*, *FBRSL1*, *GALNT9*, *MUC8*, *NOC4L*, *SNORA49*	
SRO040	13q34	Del	17.78	** *COL4A1* ***** **	Micrognathia
SRO041	15q11.2	Del/Dup	490.07	***CYFIP1********, *ELMO2P1*, *GOLGA8I*, *NIPA1*,***NIPA2********, *TUBGCP5*, *WHAMMP3*	
SRO042	16p13.11	Dup	88.17	*ABCC1*, *ABCC6*	Long philtrum
SRO043	16p11.2	Dup	214.84	*ATP2A1*,***ATXN2L********, *CD19*, *LAT*, *MIR4517*, *MIR4721*, *NFATC2IP*, *RABEP2*,***SH2B1********, *SPNS1*, *TUFM*	
SRO044	16p11.2	Del/Dup	21.6	*SPN*	
SRO045	16p11.2	Del/Dup	516.64	*ALDOA*, *ASPHD1*, *C16orf54*, *C16orf92*,***CDIPT********, *CDIPT-AS1*, *DOC2A*, *FAM57B*, *GDPD3*, *HIRIP3*, *INO80E*, *KCTD13*, *KIF22*,***MAPK3********, ***MAZ* ***, *MVP*, *PAGR1*, *PPP4C*, *PRRT2*, *QPRT*, *RN7SKP127*, *SEZ6L2*, *SPN*,***TAOK2********, *TBX6*, *TMEM219*, *YPEL3*, *ZG16*	
SRO046	16p13.3	Del/Dup	121.78	***RBFOX1********, *RNU6-457P*	
SRO047	16q24.2	Dup	99.1	***ZC3H18********, *ZFPM1*	Micropenis
SRO048	16p13.2	Del/Dup	37.53	** *GRIN2A* ** *****	
SRO049	17p13.3	Del/Dup	251.23	***CRK********, *INPP5K*, *MYO1C*,***PITPNA********, *PITPNA-AS1*, *TUSC5*,***YWHAE********	Short neck
SRO050	17q23.2	Del/Dup	314.38	*BRIP1*,***INTS2********, ***MED13* ***, *RN7SL800P*, *POLRMTP1*	
SRO051	17p13.1	Del/Dup	4.52	*-*	Low-set ears
SRO052	18p11.31	Del/Dup	429.94	***DLGAP1********, *DLGAP1-AS1*, *DLGAP1-AS2*, *IGLJCOR18*, *RN7SL39P*, *RPL21P127*, *RPL31P59*, *TGIF1*	
SRO053	22q11.1	Del/Dup	2.19	*-*	Microcephaly
SRO054	22q11.21	Del/Dup	77.94	*CLTCL1*, *KRT18P62*	Preauricular skin tag
SRO055	22q11.21	Del/Dup	8.18	*PI4KA*	Hearing impairment
SRO056	22q11.21	Del/Dup	29.74	***CRKL* ***, *RN7SL389P*	
SRO057	22q11.21	Del/Dup	13.9	*PPIL2*, *YPEL1*	Cleft palate
SRO058	22q11.22	Del/Dup	34.4	*PPM1F*, *TOP3B*	
SRO059	22q11.22	Del/Dup	85.87	*IGLV2-11*, *IGLV2-5*, *IGLV2-8*, *IGLV3-10*, *IGLV3-4*, *IGLV3-6*, *IGLV3-7*, *IGLV3-9*, *MIR650*	
SRO060	22q11.22	Del/Dup	28.54	*-*	
SRO061	22q11.22	Del/Dup	9.29	*-*	
SRO062	22q11.23	Del/Dup	1.74	***BCR* * **	
SRO063	22q12.3	Del/Dup	83.35	*RBFOX2*	
SRO064	22q13.2	Dup	502.25	*C22orf46*, *CCDC134*, *CENPM*, *CYP2D6*, *CYP2D7P*, *CYP2D8P*, *FAM109B*, *HMGN2P10*, *LINC00634*, *MEI1*, *MIR33A*, *MIR378I*, *NAGA*, *NDUFA6*, *NDUFA6-AS1*, *NHP2L1*, *OLA1P1*, *RNU6-476P*, *RNU6ATAC22P*, *SEPT3*, *SHISA8*, *SLC25A5P1*, *SMDT1*,***SREBF2********, ***TCF20* ***, *TNFRSF13C*, *WBP2NL*	
SRO065	22q13.2	Dup	69.15	*RN7SKP80*, *RNU6-513P*, *RRP7A*, *RRP7B*, *SERHL*, *SERHL2*	
SRO066	22q13.33	Del/Dup	8.67	*MAPK11*, ***PLXNB2********	
SRO067	22q13.33	Del/Dup	3.06	*NCAPH2*	Small for gestational age
SRO068	22q13.33	Del/Dup	2.01	*-*	
SRO069	22q13.33	Del/Dup	8.77	***SHANK3* * **	
SRO070	Xp11.22	Del/Dup	2.7	*-*	
SRO071	Xp11.22	Dup	0.21	*-*	
SRO072	Xp11.22	Del/Dup	7.86	*-*	
SRO073	Xp11.22	Dup	1.58	***HUWE1* * **	
SRO074	Xp22.31	Del/Dup	150.48	*MIR651*	
SRO075	Xp22.31	Del/Dup	16.6	*-*	

***** Candidate gene. ****** This column includes the most common comorbid phenotype other than neurodevelopmental abnormality. For a complete list of commonly comorbid phenotypes, please see Appendix A.

## Data Availability

Publicly available datasets were analyzed in this study. This data can be found here: https://decipher.sanger.ac.uk accessed on 26 May 2020. Data from our center presented in this study are available on request from the corresponding author.

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
