# Peer review of "Identification of Small Regions of Overlap from Copy Number Variable Regions in Patients with Hypospadias"

_ijms, 2022, doi:10.3390/ijms23084246_

Round 1

Reviewer 1 Report

The paper explores the DECIPHER database, an important online repository of genotype and phenotype data, in order to delineate genomic regions/genes which may be associated with hypospadias development.

The topic is fit for the scope of the IJMS section. However, there are major problems in the design of the study thus major revision needed.

Comments for authors:

The type of analysis used for this paper – interrogation of an extensive database and selection of patients based on a specific phenotype – has been performed previously and can bring new data. However, taking into account that DECIPHER is a database for developmental problems, hypospadias is, for a vast majority of patients, present in a complex phenotype. Thus, a simple identification of small region of overlap for hypospadias patients can lead to erroneous conclusions. Firstly, there are most probably many other common clinical features, such as developmental delay, intellectual disability, other malformations, etc, in the hypospadias patient group.  Secondly, taking into account that hypospadias is a multifactorial developmental problem, the lack of data regarding non-genetic causes represents a serious problem. In brief, spurious association can be made from such a study.  

Thus, I would suggest changing the title according to the abovementioned. The terms “candidate genes” or “chromosome loci” for hypospadias should not be used in the title or throughout the manuscript. The terms can not be used in absence of the physiological, biochemical or functional data or in absence of statistical significance for associations. Instead, the phrasing “genomic/gene structural variation observed in patients with developmental problems including hypospadias seems to better reflect the data”.

A variant for the title could be: “Identification of Small Regions of Overlap from Copy Number Variable Regions in Patients with Hypospadias”.

The abstract should also be reformulated taking into account the above comments. The introduction should be extended with the newest hypospadias genetic studies.

The methods should be clearly described with more details:

  • Patient selection: how were the DECIPHER patients selected? For exemple: Which HPO terms were used for patients call? The inclusion/exclusion criteria? Patients with 46,XX or unknown karyotype were included in the study or excluded? It should be clarified.
  • CNV evaluation: both duplications and deletions were included, but only pLi and haploinsuficienty scores are used for gene evaluation. No information regarding the triplosensitivity score for those genes located in duplicated regions is provided.
  • For the CNVs without gene content a computational interrogation for regulatory elements would be important.
  • A statistic analysis of the genes included in SRO according to their putative biological function (Gene Ontology (GO), KEGG Pathways, etc) would be useful to investigate if there is an enrichment for a specific pathway / biologic process. A particular attention should be given to the biological pathways known to be involved in hypospadias pathogenesis.

The results and discussions should be reformulated taking into account the comments above.

References: the citation of the references in the text is not in concordance with the reference list.

One of the references doesn’t seems to be be linked with the text: Theisen, J.G.; Amarillo, I.E. Creating Affirmative and Inclusive Practices When Providing Genetic and Genomic Diagnostic and Research Services to Gender-Expansive and Transgender Patients. The Journal of Applied Laboratory Medicine 2020, doi:10.1093/jalm/jfaa165.

Reviewer 2 Report

The study conducted by Dr. Scott and Dr. Amarillo has aimed to identify recurrent CNVs among individuals with isolated and syndromic hypospadias. The topic is interesting, since knowledge on the genetic background of this abnormal sexual development is still not very advance. Considering the fact that hypospadias is one of the most common types of DSD in males, identifying of causative or associated with genetic factors is very needed. The authors used the DECIPHER database (469 hypospadias cases) as well as some internal institutional databases (32 patients) to identify small recurrent CNVs.  The applied approach allowed to identify 75 regions harboring CNVs and indicate candidate genes in these regions. The list of genomic regions and genes may be useful for other research group working on genetic background of hypospadias. For this reason, I think that the work deserves publication, however, due to the very simple methodological approach, the manuscript should be prepared as a brief report.

Other comments to the manuscript:

  1. The list of references was prepared incorrectly – there are no numbers in the list and it is not easy to find a specific reference.
  2. In introduction, some information on how “smallest regions of overlap (SROs)” were defined should be added.
  3. The authors mainly work on cases of hypospadias with comorbid phenotypes – is it possible to analyze obtained results in some groups with common comorbid phenotypes, to see if identified CNVs may be associated with a set of characteristics for DSD. Now the data are shown only in supplementary tables. Maybe some results of this research could be presented in the main text as the hallmark of this work is the analysis of hypospadias cases with other congenital comorbidities. It would be more beneficial for readers than showing two times data about the localization of CNVs on chromosomes (Fig. 2 and 3)
  4. The discussion part is too long and should be limited only to these fragments which concerns discussion with obtained results. For example, in the subsection “Hypospadias and Sex Chromosomes” there are 3 paragraphs about other issues.

Round 2

Reviewer 1 Report

The authors took in consideration the comments and made the necessary changes to the manuscript. Although, the “candidate gene” term is maintained in the manuscript, the association with hypospadias is more cautiously presented.

A minor change is  still necessary in abstract:

Please use the term ”potential” candidate genes in the abstract  (line18-19).

Thus, the revised version of the manuscript can be publish.

Author Response

Thank you for reviewing our manuscript again; we appreciate your time and effort. We will make this adjustment and resubmit.